# Synthesis of cationic liposome nanoparticles using a thin film dispersed hydration and extrusion method

Alessandro Cazzolla[1,2,3]*, Julie Rose Mae Mondala[1,2,3], Janith Wanigasekara[1,2,3,4], Joanna Carroll[1], Noah Daly[1], Brijesh Tiwari[4], Alan Casey[5], James F. Curtin[2,3,6]*

1 School of Food Science and Environmental Health, Technological University Dublin, Dublin, Ireland, 2 Environmental Sustainability & Health Institute (ESHI), Greenway Hub, Technological University Dublin, Dublin, Ireland, 3 FOCAS Research Institute, Technological University Dublin, Dublin, Ireland, 4 Department of Food Biosciences, Teagasc Food Research Centre, Ashtown, Dublin, Ireland, 5 School of Physics, Clinical and Optometric Sciences, Technological University Dublin, Dublin, Ireland, 6 Faculty of Engineering & Built Environment, Technological University Dublin, Dublin, Ireland

* james.curtin@tudublin.ie (JFC); alessandro.cazzolla@tudublin.ie (AC)

**Data Availability Statement:** All relevant datasets that support the findings of this study are uploaded into OSF and can be accessed using the following DOI: www.doi.org/10.17605/OSF.IO/TYGZ9.

## Abstract

Liposome nanoparticles can carry a wide range of therapeutic molecules including small molecules and nucleic acid-based therapeutics. Potential benefits include translocation across physiological barriers, reduced systemic toxicity, and enhanced pharmacokinetic parameters such as absorption, distribution, selective release and optimal elimination kinetics. Liposome nanoparticles can be generated with a wide range of natural and synthetic lipid-based molecules that confer desirable properties depending on the desired therapeutic application Nel et al (2023), Large (2021), Elkhoury (2020). This protocol article seeks to detail the procedures involved in the production of cationic liposomes using thin-film dispersed hydration method with an estimated uniform size of 60–70 nm for targeted drug administration in tumor cells, by modifying the previous one also published by the same authors cited here. The method was carrying out using N-[1-(2,3-dioleoyloxy)propyl]-N,N,N-trimethylammonium methyl (DOTAP, 2 mg) as cationic lipid and cholesterol (0.5 mg) in a molar ratio of 7:3 respectively. The liposomal suspension was obtained and its physical, chemical and biological properties were determined. A two-step extrusion process, using 100 nm and 50 nm polycarbonate membranes, was carried. The results demonstrate generation of liposome nanoparticles with a size of 60–70 nm stable for at least 16 weeks and with an encapsulation efficiency of approximately 81% using Doxorubicin.

## 1. Introduction

Nanoparticles represent the future of targeted and sustained drug delivery, with huge implications for addressing and treating complex diseases such as cancer [1, 2]. These nanovesicles can be turned into intelligent systems capable of encapsulating therapeutic and imaging compounds while remaining stealthy. Manipulation of the size, surface characteristics, and

**Funding:** This study was funded by a TU Dublin Research Scholarship (AC) www.tudublin.ie and by Science Foundation Ireland, 21/FFP-A/9189 (JC) www.sfi.ie.

**Competing interests:** The authors declare no competing interests.

**Abbreviations:** DOTAP, 1,2-dioleoyl-3-trimethylammonium-propane; DSPC, 1,2-distearoyl-sn-glycero-3-phosphocholine; DOXO, Doxorubicin; PdI, Polydispersity index; PI, Propidium Iodide; Tc, Lipid's transition temperature.

material composition of these new systems can aid in drug delivery and controlled release therapy. Of all the indications or uses of nanoparticles for therapeutic purposes in clinical trials, cancer nanomedicines receive the greatest attention. The foundation of this interest is the track record and efficacy of licensed nanomedicines like Abraxane and Doxil, which taken together account for most of all nanoparticle treatments presently undergoing clinical trials [3].

Most of these systems are liposome-based, just like nanoparticles being studied for other indications and uses. Lipid nanoparticles, despite their own discovery some 60 years ago, they prove to be widely used nowadays in the area of drug delivery as potentially viable and flexible drug vesicles. They can outperform traditional drug delivery systems in terms of site-targeting, sustained or controlled release, drug protection from degradation and clearance, therapeutic efficacy, and hazardous side effects [4–6]. Among nanoparticles, liposomes, have been identified as potentially viable and flexible drug delivery systems. Discovered in 1965 by Alec D. Bangham, these are lipid vesicles that form approximately spherical unilamellar or multilamellar bilayer shapes with an aqueous centre [4–6]. When amphiphilic lipids, such as phospholipids, are spread in water, these vesicles develop spontaneously and closely mimic our biological membranes [4, 7]. In terms of biocompatibility and biodegradability, this similarity is advantageous for drug delivery. Liposomal vesicles' shape allows for the encapsulation of a wide range of drugs with lipophilic and/or hydrophilic properties, with hydrophobic/lipophilic and amphiphilic molecules incorporated into the bilayer membrane and hydrophilic compounds confined in the aqueous compartments [4, 8]. Depending on the type of lipid, or the lipid composition used, various properties can be influenced, including the fluidity of the bilayer, as well as the surface charge and method of preparation of the liposomes themselves. The toxicity of cationic lipids, which can activate a number of cellular pathways including pro-apoptotic and pro-inflammatory cascades, remains an issue and is one of the main obstacles to their use [9]. Phospholipids, which can be natural or synthetic in origin, are the most widely utilized lipids, however they have a drastically shortened shelf life and a limited ability to safeguard encapsulated compounds due to their high permeability, which leads to drug leakage. To prevent this, sterols, primarily cholesterol, should adjust membrane stiffness and stability [10, 11]. When liposomes encapsulate substances, their ultimate goal is to protect them from their surroundings, shielding them from enzymatic and chemical activities involving pH or temperature changes (Fig 1A) [4, 12]. To improve the properties of liposomes, allow for longer retention within the body, and make them more specific for a particular application, surface modifications can be performed, introducing PEG chains or particular functional groups to maximize affinity for the target molecule (Fig 1B and 1C) [13]. In presence of multiple targeting ligands but also imaging or fluorescent agents within the liposome, we call them multifunctional liposomes (Fig 1D) [4, 14].

Taking all of these qualities into account, liposomes provide several distinct advantages as a drug delivery technology, including the capacity to self-assemble, load hydrophilic, hydrophobic, and amphiphilic molecules, improve solubility, protect the encapsulated drugs, provide biocompatibility, target specificity and low toxicity at relative levels, biodegrade, and cause low immunogenicity [4, 5, 15]. Liposome size has an important impact in the successful transport of anticancer medicines to the tumor location. It has been shown to influence its time in circulation, tumor accumulation, tumor retention, and medication release. According to available evidence, a liposome diameter of less than 200 nm is the optimal size for effective drug administration, particularly when targeting the brain and crossing the blood brain barrier [16]. However, research on using liposomes therapeutically to deliver drugs specifically to the brain is still in its infancy. Nonetheless, a number of licensed liposomal drugs have received approval for use in clinical settings, and ongoing studies are examining their potential. Ananda et al.

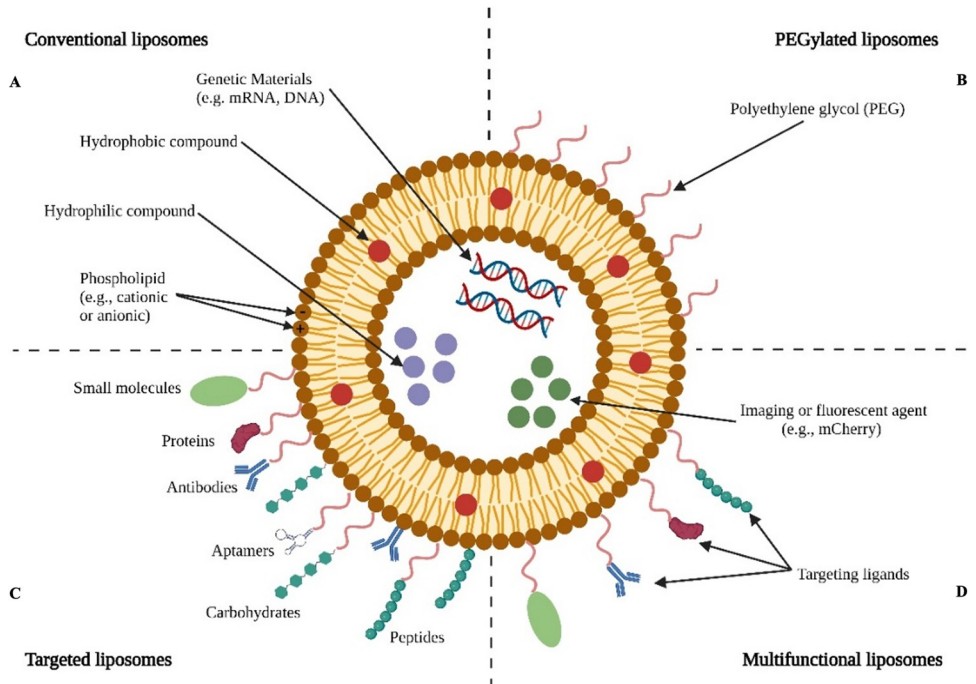

**Fig 1. Representation of liposomes.** Representation of Conventional, PEGylated, targeted and multifunctional liposomes (modified from Nel et al. Bioactive Materials 2023 [4]. Figure created with BioRender).

studied the effects of Temozolomide and Doxil® on 40 patients with GBM in a phase 2 trial. The median time to progression was 6.2 months, and the median progression-free survival (PFS) was 13.4 months. Twenty-eight patients had stable disease, five showed disease progression, and one patient showed a full response [17].

Doxorubicin (DOXO) is one of the most widely recognized and commonly used chemotherapeutic medications for the treatment of numerous forms of cancer because of its FDA approval and multifunctionalization, yet its usage is challenging due to its severe cardiotoxicity. These negative consequences have propelled researchers to create alternative drug delivery systems that use nanotechnology to transport DOXO [18]. Recent studies highlight that DOXO is used in combination with liposomes as targeted chemotherapy delivery to non-resectable primary pancreatic tumours [19].

In this study, we used thin-film dispersed hydration method to obtain lipid nanoparticles. This method resulted to be robust, reproducible, simple and easily repeated. Double membrane extrusion method was used to obtain the size of desired lipid nanoparticles with a good polydispersity index. Stability was also studied in terms of size and zeta potential. Doxorubicin encapsulation efficiency % was obtained through absorbance analysis and cytotoxicity was evaluated in U-251MG, U-87 and A-172 human glioblastoma (GBM) cell lines.

## 2. Methodology

The part of the protocol described in this peer-reviewed article "Synthesis of cationic liposome nanoparticles using a thin film dispersed hydration and extrusion method" is published on protocols.io, https://dx.doi.org/10.17504/protocols.io.dm6gpjyx1gzp/v2 and is included for printing as S1 File with this article.

## 2.1 Ethics statement

The research project was approved by TU Dublin Research Ethics and Integrity Committee and involved the use of human samples. Human cancer cell lines (U-251 MG, U87 MG and A-172) were used in the study. These are cell lines obtained from reputable commercial cell banks, these are established, commercially available cell lines and consent was not obtained from the original donors. Animal tissue (fetal calf serum) was also used in the study. This was obtained from a reputable commercial company. This article does not contain any studies with human participants or animals performed by any of the authors. All research and scholarly activities were reviewed and approved by the TU Dublin's Research Ethics and Integrity Committee (REIC) in accordance with the TU Dublin Code of Conduct.

## 2.2 Chemicals

All chemicals used in this study were supplied by Sigma-Aldrich—Merck Group unless stated otherwise.

## 2.3 Liposomes synthesis

Liposomes were prepared with a combination of 1,2-dioleoyl-3-trimethylammonium-propane (DOTAP) or 1,2-distearoyl-sn-glycero-3-phosphocholine (DSPC) and cholesterol via a modification of a thin-film dispersed hydration method reported by A. Yusuf in 2017 [20, 21]. DOTAP (2 mg) and cholesterol (0.5 mg) were weighed respectively in a 7:3 molar ratio, dissolved in a fixed amount of chloroform (5 mL) and stirred for 15 minutes above the lipid's transition temperature, $T_c$ (40˚C). The resulting solution was placed in a vacuum oven set at 40˚C to let the chloroform to evaporate. The lipid film (also called lipid cake) was further dried by incubating overnight in an oven set at just above $T_c$. The lipid film was rehydrated in ultrapure water to make the final concentration of 1 mg/mL (2.5 mL were added). Encapsulation of DOXO is made passively by rehydration with a solution 0.006 mg/mL DOXO in ultrapure water (concentration 1/100 w/w than the lipid). The solution was stirred for 30 minutes above lipid $T_c$ and vortexed for 2 extra minutes. After vortexing, the solution was stored at 4˚C overnight (Fig 2).

This is to prevent acid and base hydrolysis reactions taking place in lipids with ester-linked hydrocarbon chains. Temperature has a significant impact on the rate of reaction so the solution should be refrigerated to slow this process. In this study the goal was to obtained liposomes < 100nm. Hence, to achieve this, an extrusion using a Micro Extruder set (Avanti, Polar Lipids) was carried out. A two-step extrusion process, using 100 nm and 50 nm polycarbonate membranes, was carried out above the $T_c$ of the lipid. Extruded liposomes were stored in 4˚C to maintain liposome stability. The suspension was subjected to dynamic light scattering and zeta potential analysis for size and surface charge measurements respectively.

## 2.4 Characterization of synthesized liposomes

Liposome properties, such as particle size, polydispersity index and zeta potential are important parameters to study when synthesising liposomes for drug delivery. Dynamic light scattering and zeta potential tests are now widely used as quick, straightforward, and repeatable methods to determine particle size and the charge developed at the interface between a solid surface and its liquid medium. The first is a well-known and accurate measurement method for identifying particle sizes in suspensions and emulsions. Its foundation is the Brownian motion of particles, which posits that in a liquid, smaller particles move more quickly and larger ones move more slowly. The information about the diffusion speed and, consequently,

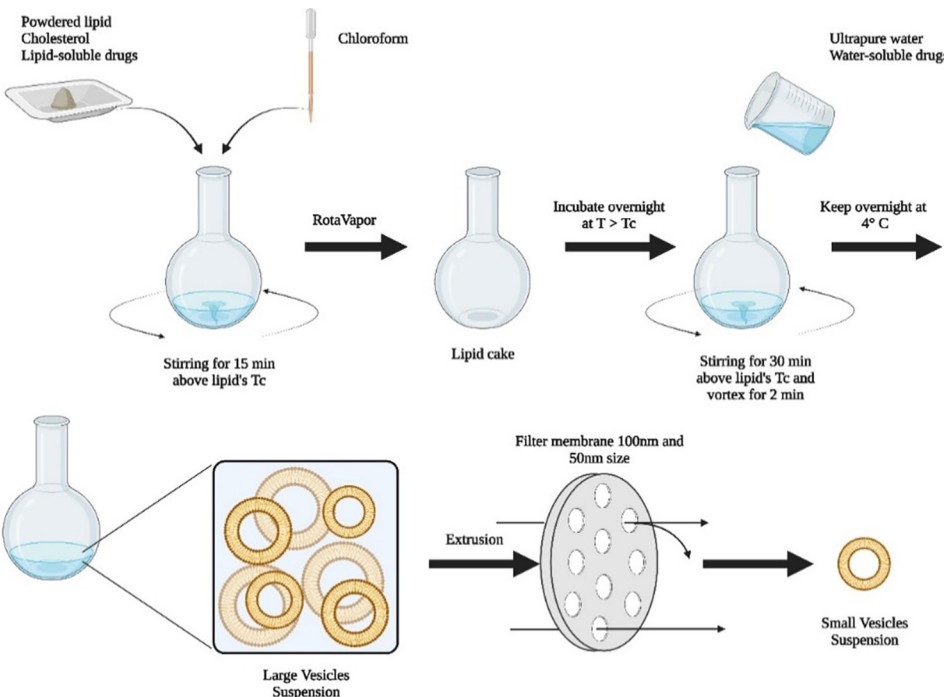

**Fig 2. Thin-film dispersed hydration method.** Synthesis step by step of liposomes via a thin-film dispersed hydration method (Figure created with BioRender).

the size distribution, is included in the light scattered by the particles. The diffusion coefficients ($D_t$) of the particles are inversely proportional to the size (hydrodynamic radius ($R_H$)) of the particles according to the Stokes-Einstein relationship.

$$D_t = \frac{k_g T}{6\pi\eta R_H}$$

$k_g$ = Boltzmann constant ($1.38064852 \times 10^{-23}$ J/K), T = temperature, $\eta$ = absolute viscosity and RH = hydrodynamic radius [22, 23]. All the results were carried out with Malvern Zetasizer Nano ZS (Malvern Panalytical, Malvern, UK). According to the number of peaks obtained or the PdI values is also possible to determine if liposomes are multilamellar or unilamellar the zeta potential measurement of the nanoparticles was also measured with Malvern Zetasizer Nano ZS instrument set at 25°C for all the samples. Nanoparticles were loaded into a disposable folded capillary cell up to the marked portion.

Solvent Water (Viscosity 0.8872, RI 1.330)

Material DOTAP (RI 0.149, Absorption 0.01)

Zeta–Smoluchoski Model F(ka) 1.50

Scanning electron microscope (SEM)—Used to observe the liposomes morphology.

## 2.5 2D cell culture

The human glioblastoma cell line U-251 MG (formerly known as U-373MG-CD14) was a gift from Michael Carty (Trinity College Dublin) meanwhile A-172 and U-87 human brain GBM cell lines were a gift from Dr. Brona Murphy (Royal College of Surgeons in Ireland). The absence of mycoplasma was checked by using MycoAlert® Mycoplasma Detection Assays (Lonza). Cells were maintained in Dulbecco's modified Eagle medium (DMEM)-high glucose

supplemented with 10% fetal bovine serum (FBS) and 1% penicillin/streptomycin. Cells were maintained in a humidified incubator containing 5% $CO_2$ atmosphere at 37°C in a TC flask T75, standard for adherent cells (Sarstedt). Cells were routinely subcultured when 80% confluence was reached using 0.25% w/v Trypsin solution.

## 2.6 Cell viability assay

Cell viability was analysed using AlamarBlue™ cell viability reagent (Thermo Fisher Scientific). U-251 MG, A-172 and U-87 cells were seeded at a density of $2 \times 10^3$ cells/well overlaid with 100 µl cell culture medium/well into flat-bottom 96-well plates (Sarstedt, Ltd.). Cells were incubated overnight at 37°C in a humidified atmosphere. After 24h incubation, media was removed and each well was treated with 100 µL liposomes or liposome encapsulated solution in serial dilution starting from 0.4 mg/mL as higher concentration. Dimethyl sulfoxide (DMSO) (20%) was used as positive control and media as negative control. Cell viability assays tests were conducted also using solutions of DOTAP:Cholesterol and DSPC: Cholesterol, with same molar ratio (7:3 respectively) then in liposomes, in 0.5% of DMSO to understand if the materials themselves are harmful for the cells and at which concentration.

## 2.7 Encapsulation of doxorubicin

A standard absorbance curve was plotted starting from a concentration of 0.02 mg/mL DOXO in ultrapure water. By knowing the initial concentration of the DOXO solution used in the synthesis steps and determining how much of it is present in the supernatant subsequently the purification process the percentage of encapsulation efficiency (EE) can be determined as follows:

$$EE\ \% = \frac{DOXO\ starting\ concentration - free\ DOXO\ concentration}{DOXO\ starting\ concentration}\ x\ 100$$

Before analysing the encapsulation by absorbance analysis, a pre-separation was carried out by dynamic light scattering analysis to understand the best filter to use and whether to extrude pre- or post-centrifugation. Amicon® Ultra-4 Centrifugal Filter Units of 50kDa and 10kDa were used to purify liposomes and the presence or absence of nanoparticles in the supernatant was confirmed by dynamic light scattering analysis. Liposomes were encapsulated with doxorubicin (DOXO), concentration of 0.006 mg/mL in ultrapure water, and absorbance tests were carried out using maximum peak as excitation wavelength. The chosen Amicon® Ultra-4 Centrifugal Filter Unit was then used to separate free DOXO from encapsulated DOXO. The aim would be to retain the nanoparticles at the top, allowing anything that has not been encapsulated to pass through the filter, which is dissolved in the solution.

## 2.8 Statistical analysis

At least three independent replications of each experiment were performed. The data are displayed as mean ± average standard deviation. Curve fitting and statistical analysis were performed using Prism 9.1.0, GraphPad Software, Inc. Nonlinear regression was used to measure the dose-response curves. All figures have error bars that were displayed using the mean of the standard errors or the standard error of the mean, depending on the result, and data are provided with their own unit or as percentages.

## 3. Expected results and discussion

### 3.1 Synthesis

When synthesizing liposomes, the thin layer method, which starts with the desired formulation and uses cloroform as a solvent, is very popular. The results listed in Fig 3A and the images obtained using Scanning Electron Microscope (SEM, Fig 3B) demonstrate the effectiveness of the theory. The average size of the liposomes in terms of intensity is 640.9 ± 34.0 nm, and its PdI value is 0.42 ± 0.11. Instead, taking into account the average size in terms of numbers, the outcome is 479.5 ± 27.2 nm. In this section average size results are reported as mean value ± 25% of the range (Interquartile range IQR) [5].

### 3.2 Extrusion study passes

The objective was to obtain liposomes that were less than 100 nm in size, and to do this, an extrusion implementing a Micro Extruder set (Avanti, Polar Lipids) was conducted. The protocol provided by Avanti was amended by including a double extrusion with a study performed to determine the appropriate number of steps to take. Two different membranes made of polycarbonate were used, one each at 100 nm and 50 nm. Passes from 5 to 17 were completed with each membrane (resulting in samples generated with between 10 and 34 total

| | DOTAP No Extrusion | | | | |
| | Size Intensity [nm] | STD Dev | Polydispersity index (PdI) | Size Number [nm] | STD Dev |
|---|---|---|---|---|---|
| | 665.5 | 138.9 | 0.26 | 617.4 | 147.1 |
| | 699.1 | 154.9 | 0.34 | 655.3 | 159.1 |
| A | 705.1 | 163.3 | 0.54 | 654.8 | 166.9 |
| | 577.0 | 112.2 | 0.40 | 577.0 | 112.2 |
| | 548.0 | 101.3 | 0.36 | 548.0 | 101.3 |
| | 555.0 | 106.5 | 0.42 | 527.0 | 116.1 |
| | 706.3 | 158.3 | 0.36 | 660.2 | 162.6 |
| | 656.7 | 148.8 | 0.61 | 39.7 | 6.6 |
| | 655.1 | 139.1 | 0.50 | 36.0 | 5.7 |
| Average | 640.9 | 135.9 | 0.42 | 479.5 | 108.6 |
| STD Dev | | | 0.11 | | |
| Results | 640.9 ± 135.9 | | 0.42 ± 0.11 | 479.5 ± 108.6 | |

B

SU6600 20.0kV 5.4mm x6.00k SE 5.00um

**Fig 3. Physical properties of liposomes.** Analysis by dynamic light scattering of liposomal particles, subsequent to synthesis (A). Scanning Electron Microscope image of liposomes (B).

passes through the extruder) to observe any differences in average size, polydispersity index, and zeta potential in order to determine the ideal value. The number of extrusions is always odd for each membrane, because material that cannot cross the membrane are retained in the first syringe and can thereby thicken the resultant solution for even numbers of passes. Nanoparticles with a zeta potential between—10 and + 10 mV are considered approximately neutral, while nanoparticles with zeta potentials of greater than + 30 mV or less than—30 mV are considered strongly cationic and strongly anionic, respectively. A large positive or negative value of zeta potential above ± 30 mV is considered to indicate good physical colloidal stability. On the other hand, values less than ± 30 mV can result in particle aggregation, flocculation, and precipitation due to the van der Waal forces of attraction which result in physical instability of the colloidal suspension [23]. This value, representing repulsion forces, represent only a small part of the much larger picture. There would be forces of attraction to consider as well, but this value was just given as an indicator of the stability of the nanoparticle suspension. The observed potential Z is 59.4 ± 12.4 as shown below, which validates the proper use of the cationic lipid (DOTAP) and suggests a certain level of stability (Fig 4A). The average standard deviation was found to have very low interexperiment variability. As can be seen from Fig 4A and 4B there is a clear shrinkage of most of the liposomes, starting from about 500 nm down to 60–70 nm, confirming the effectiveness of the amended extrusion method. A greater number of extrusions makes it possible to obtain smaller and smaller particles, 61.5 ± 16.2 nm for 17 passes, although with a nearly identical polydispersity index, always around 0.07. The same thing, however, cannot be said at the zeta potential level. In fact, the results obtained suggest how a greater number of passes, hence a smaller nanoparticle, lead to a smaller zeta potential, 26.7 ± 15.6 mV for 17 passes.

Since the zeta potential is an important figure to avoid the formation of agglomerates in the solution (> + 30 mV), 7 (14 total passes considering the two membranes) passes was chosen as the optimal value. It shows a final value of average size of the liposomes in terms of intensity is 103.4 ± 30.9 nm, and its PdI value is 0.07 ± 0.01. Instead, considering the average size in terms of numbers, the outcome is 69.6 ± 18.8 nm with a zeta potential of 47.9 ± 24.1 mV. From the SEM image, Fig 4C, we can observe how effectively the nanoparticles, subsequent to extrusion, fall perfectly within the range of values obtained by Dynamic light scattering. Here it can be seen that indeed a very good percentage of the nanoparticles are actually below 100 nm, but not all of them. An observation must be made in terms of intensity zeta size and number zeta size. The quantity of light scattered by the particles in the various size bins is indicated by the intensity distribution. The number distribution shows how many particles there are in each of the different size bins. The former value turns out to be always higher than the latter since large particles scatter much more light than small particles, the intensity of scattering of a particle is proportional to the sixth power of its diameter (from Rayleigh's approximation) [22]. This in turn explains that while liposomes are physically smaller on average, there are larger liposomes in the mixture that increase the final intensity value.

### 3.3 Stability COLD and ROOM temperature

The stability of the nanoparticles in terms of size, polydispersity index and zeta potential was analysed over time considering three different time points, 1 week, 4 weeks and 16 weeks post synthesis. The encapsulation of Propidium Iodide (PI) was carried out in the second step of the synthesis, as explained by protocol (1/20 w/w of the lipid) [24]. Three different sets of samples were prepared for analysis, DOTAP, DOTAP-PI stored at 4°C and DOTAP, DOTAP-PI stored at room temperature (Fig 5A and 5B). Results in terms of size will be discussed considering the number distribution. In the case of nanoparticles stored at low temperature their size

| Samples | Size [nm] | STD dev [nm] | PdI | STD Dev | Mean Peak [nm] | STD dev [nm] | Zeta [mV] | STD dev [mV] |
|---------|-----------|--------------|------|---------|----------------|--------------|-----------|--------------|
| 0 | 640.9 | 135.9 | 0.42 | 0.11 | 479.5 | 108.6 | 59.4 | 12.4 |
| 5 | 106.9 | 32.8 | 0.08 | 0.02 | 70.9 | 19.4 | 44.1 | 20.4 |
| 7 | 103.4 | 30.9 | 0.07 | 0.01 | 69.6 | 18.8 | 47.9 | 24.1 |
| 9 | 97.4 | 28.7 | 0.07 | 0.01 | 66.3 | 17.4 | 39.3 | 14.7 |
| 11 | 99.5 | 29.9 | 0.08 | 0.01 | 66.9 | 17.8 | 23.8 | 23.0 |
| 13 | 96.2 | 29.3 | 0.08 | 0.01 | 63.4 | 17.2 | 36.6 | 15.2 |
| 15 | 94.5 | 27.2 | 0.07 | 0.01 | 65.2 | 16.9 | 37.0 | 16.4 |
| 17 | 91.5 | 27.1 | 0.08 | 0.01 | 61.5 | 16.2 | 26.7 | 15.6 |

**Fig 4. Liposome extrusion study.** Extrusion study from 7 to 17 passes: values obtained by dynamic light scattering analysis and compared by bar chart (A and B). Scanning Electron Microscope image of the chosen liposomes, 7 passes, with nanoparticle measurements in nm (C).

went from 57.5 ± 15.2 nm after 1 week to 56.3 ± 16.7 nm after 16 weeks, with polydispersity index from 0.08 ± 0.01 to 0.25 ± 0.02 in the unencapsulated case and from 63.7 ± 18.0 nm after 1 week to 63.4 ± 19.8 nm after 16 weeks, with polydispersity index from 0.10 ± 0.01 to 0.14 ± 0.01 in the encapsulated case. Zeta potential was increasing in value, but also in standard deviation, going from 7.9 ± 14.4 mV to 13.5 ± 21.5 mV in the unencapsulated case and from 18.3 ± 19.0 mV to 27.0 ± 31.3 mV in the encapsulated case.

Regarding the room temperature test, there is a marked increase in nanoparticle size from 55.5 ± 16.0 nm after 1 week to 901.1 ± 201.3 nm after 16 weeks, with polydispersity index from 0.11 ± 0. 01 to 0.43 ± 0.25 in the unencapsulated case and from 60.6 ± 19.9 nm after 1 week to 143.0 ± 95.8 nm after 16 weeks, with polydispersity index from 0.16 ± 0.01 to 0.51 ± 0.10 in the encapsulated case. In contrast to the previous case, the zeta potential value decreases at room

| | Size [nm] | | | PdI | | | Zeta [mV] | |
|---|---|---|---|---|---|---|---|---|
| | 1w | 4w | 16w | 1w | 4w | 16w | 1w | 16w |
| **DOTAP Cold** | 57.5 ± 15.2 | 54.3 ± 16.1 | 56.3 ± 16.7 | 0.08 ± 0.01 | 0.16 ± 0.02 | 0.25 ± 0.02 | 7.9 ± 14.4 | 13.5 ± 21.5 |
| **DOTAP-PI Cold** | 63.7 ± 18.0 | 54.3 ± 16.1 | 63.4 ± 19.8 | 0.10 ± 0.01 | 0.10 ± 0.01 | 0.14 ± 0.01 | 18.3 ± 19.0 | 27.0 ± 31.3 |
| **DOTAP Room** | 55.5 ± 16.0 | 52.7 ± 15.4 | 901.1 ± 201.3 | 0.11 ± 0.01 | 0.13 ± 0.01 | 0.43 ± 0.25 | 15.9 ± 13.6 | -3.3 ± 3.9 |
| **DOTAP-PI Room** | 60.6 ± 19.9 | 61.0 ± 20.9 | 143.0 ± 95.8 | 0.16 ± 0.01 | 0.17 ± 0.01 | 0.51 ± 0.10 | 39.9 ± 10.6 | 14.4 ± 11.1 |

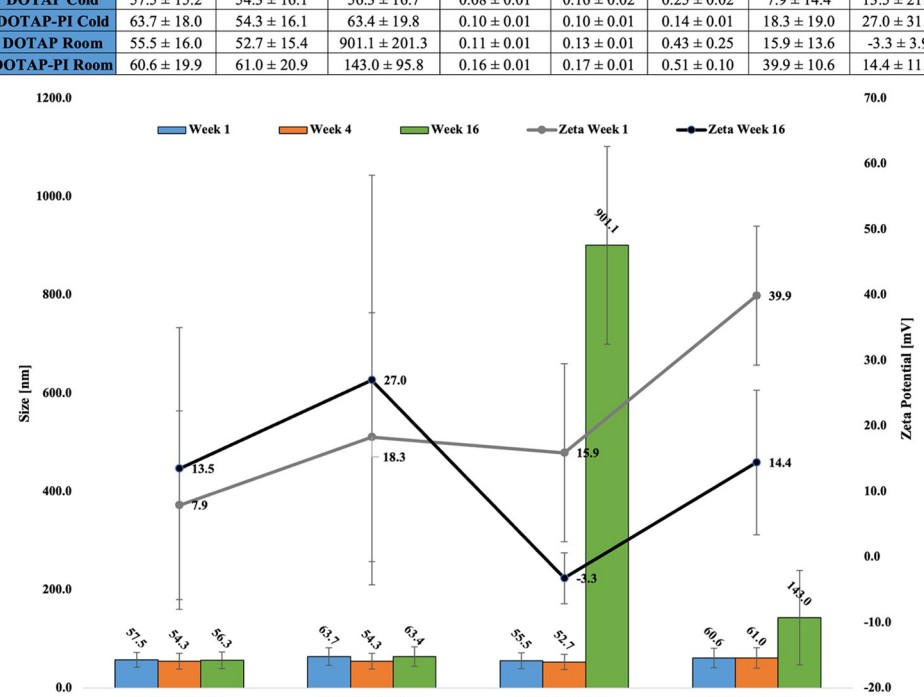

**Fig 5. Stability of liposomes.** Summary of size, polydispersity index and zeta potential through the weeks in table (A) and bar diagram (B).

temperature going from 15.9 ± 13.6 mV to -3.3 ± 3.9 mV in the no encapsulated case and from 39.9 ± 10.6 mV to 14.4 ± 11.1 mV in the encapsulated case. Further studies will be conducted to understand better the reason why of the drop of zeta potential from week 0 to week 1. The increase in terms of size for the liposomes can be explained looking at the zeta potential values. A lower value than the suggested limit suggests the formation of agglomerates. This can explain an increase of the nanoparticles size and polydispersity index Considering nanoparticles stored at low temperatures, this increase is slight, affirming excellent stability even after four months. The same result can be observed for both encapsulated and un-encapsulated mixtures. At room temperature the zeta potential after 16 weeks decreases greatly, even reaching negative values or at least around zero in the case of unencapsulated. Such low values of zeta potential demonstrate the large increase in the diameter of nanoparticles. The results show that the best storage temperature is the colder one (4°C) for DOTAP and DOTAP-PI. To demonstrate the versatility of the protocol used, the same analysis was carried out using a lipid of zwitterionic nature, DSPC, and the results ensured the stability of the liposomes obtained after the 16 weeks, confirming the robustness of the method use.

### 3.4 Encapsulation efficiency of doxorubicin

The encapsulation efficiency of the lipid nanoparticle at approximately 60–70 nm in size was determined using a drug–doxorubicin. Ultracentrifugation technique was performed to separate encapsulated and unencapsulated DOXO. To determine the most suitable filter for the liposomes under study, dynamic light scattering analyses were performed to determine the correct pore size; and to determine whether to purify the solution before or after the extrusion

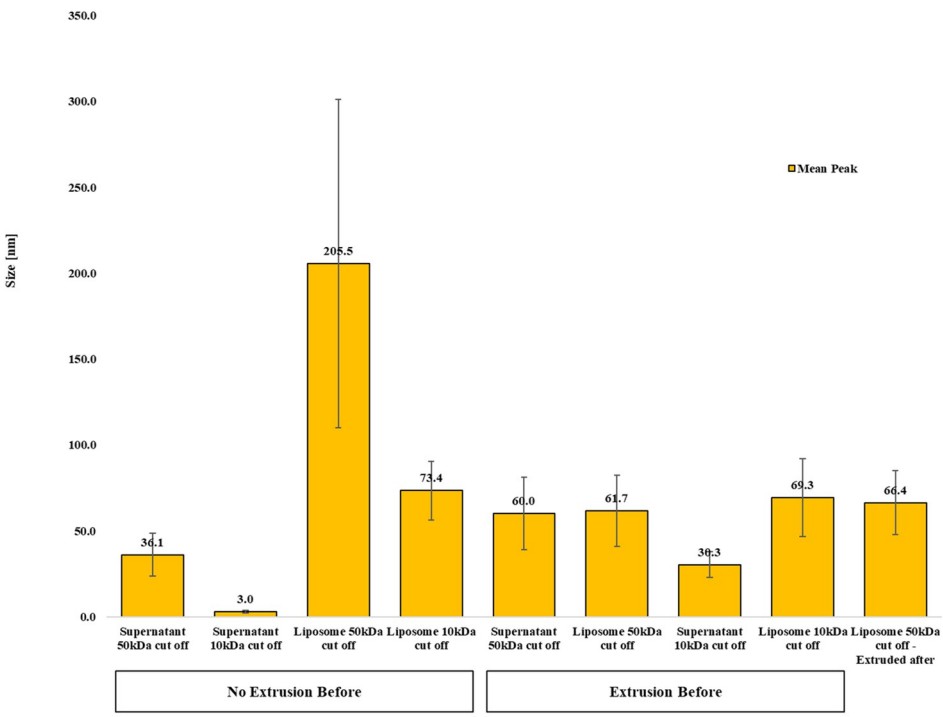

**Fig 6. Physical properties of ultracentrifuged liposomes.** Dynamic light scattering analysis of ultracentrifuged liposomes using two different molecular weight filters, 10 kDa and 50 kDa. Both supernatant and recollected liposomes where analysed.

steps. The aim would be to retain the nanoparticles at the top, allowing anything that has not been encapsulated to pass through the filter, which then floats in the solution. As can be seen from the results there is presence of liposomes in the supernatant both using the 50 and 10 kDa filters if this step is performed subsequent to extrusion. The filter chosen is the 50 kDa filter because it presents a situation consonant with what was previously discovered with the study of extrusion steps, with nanoparticles with a size greater than 200 nm that can subsequently undergo extrusion, reducing to 66.4 nm (Fig 6).

A standard absorbance curve was plotted starting from a concentration of 0.006 mg/mL DOXO in ultrapure water. The obtained equation $y = 29.537x - 0.0226$ (with $R^2 = 0.9917$) will be used to determine the concentration in the case of liposomes. By knowing the initial concentration of the DOXO solution used in the synthesis steps and determining how much of it is not encapsulated in liposomes subsequent to ultracentrifugation the percentage of encapsulation efficiency can be determined. Three different batches of DOTAP-DOXO were made and centrifuged. The absorbance values of the three supernatants are reported in the table below (Table 1) and the average was used to determine the EE% following the equation obtained above. The final result can be expressed as 80.98% ± 1.81%, confirming an excellent ability to encapsulate nanoparticles using this protocol.

### 3.5 Cytotoxicity

Cytotoxicity tests were carried out for encapsulated and unencapsulated nanoparticles using three different glioblastoma cell lines. Freshly prepared unencapsulated liposomes exhibit very

**Table 1. Absorbance and EE% values for the three supernatant solutions.**

| | Absorbance Supernatant 1 | Absorbance Supernatant 2 | Absorbance Supernatant 3 | Starting DOXO |
|---|---|---|---|---|
| | 0.006 | 0.002 | 0.009 | 0.166 |
| | 0.006 | 0.003 | 0.009 | 0.166 |
| | 0.008 | 0.003 | 0.008 | 0.166 |
| | 0.006 | 0.002 | 0.009 | 0.164 |
| | 0.007 | 0.002 | 0.01 | 0.165 |
| Average | 0.0066 | 0.0024 | 0.009 | 0.1654 |
| Concentration [mg/mL] | 0.0012 | 0.0011 | 0.0013 | 0.0064 |
| Encapsulation Efficiency % | 80.59 | 83.38 | 78.99 | |

Absorbance values of three DOTAP-DOXO supernatants after purification step. EE% was calculated by using the equation expressed above.

similar IC50 in these cell lines, around 20–25 μg/mL (Fig 7A). Next, the same nanoparticles used for the stability study (DOTAP, DOTAP-PI cold and room temperature) were tested on the U251MG cell line. The analysis was performed after 1 week and 16 weeks from the synthesis, respectively. DOTAP and DOTAP-PI liposomes exhibit similar behaviour one week after synthesis, both when stored at room temperature and 4˚C (Fig 7B and 7C).

The results show that all liposomes are more harmful to this cell line after 16 weeks. DOTAP and DOTAP-PI at room temperature turns out to be more cytotoxic than those stored at cold temperature, further emphasizing how storage temperature affects nanoparticles. This increasing in the cytotoxicity should not be related with the increase in term of size because smaller nanoparticles have greater available surface area to interact with biological constituents such nucleic acids, proteins, fatty acids, and carbohydrates due to their bigger specific surface area (SSA). Due to its smaller size, it is also probably possible to enter cells and harm them [25]. Subsequent tests should be conducted as a part of a wider study on the efficacy and safety

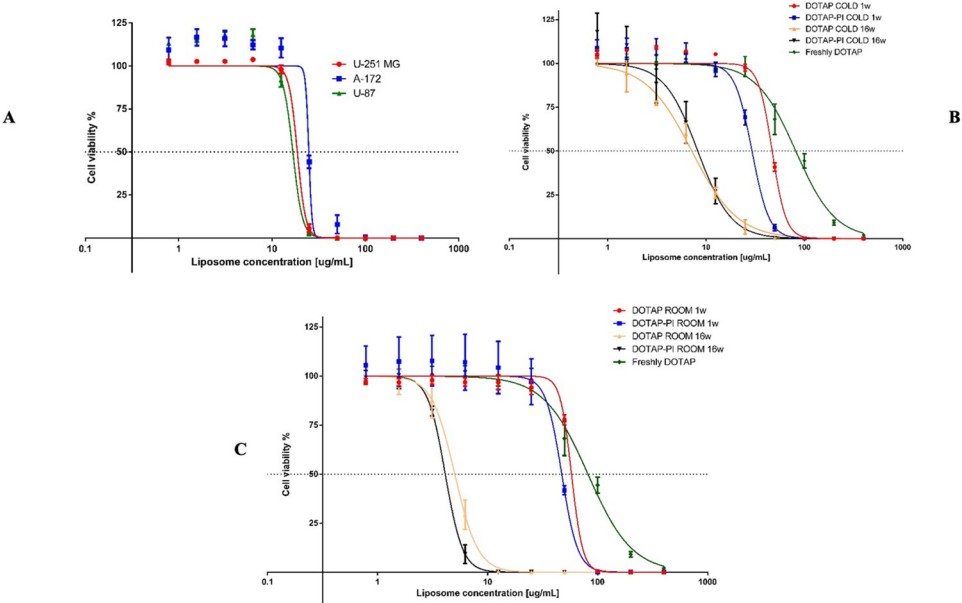

**Fig 7. Cytotoxicity of liposomes.** Cytotoxicity of plain liposomes with three different glioblastoma cell lines (A). Cytotoxicity of encapsulated PI and unencapsulated liposomes with U-251 MG through the weeks (B and C).

of the system, using an array of appropriate in vitro 2D, 3D and in vivo models. The success of the synthesis was demonstrated, with the nanoparticles obtained post-extrusion, considering 7 as the number of passages chosen, with a diameter of 69.6 ± 18.8 nm. The storage temperature appears to be a fundamental parameter concerning the stability of the nanoparticles. Storage at 4˚C appears to be the best, with stability maintained over 16 weeks. The passage of weeks appears to cause an increase in the cytotoxicity of the liposomes for U-251 MG cells, but further investigation must be conducted to understand the nature of it. The efficacy of the encapsulation of doxorubicin was determined, resulting in a final percentage of 80.98% ± 1.81%.

## Supporting information

**S1 File. Synthesis of cationic liposome nanoparticles.** Protocol published on protocols.io regarding Synthesis of cationic liposome nanoparticles using a thin film dispersed hydration and extrusion method.
(PDF)

## Acknowledgments

The authors wish to thank TU Dublin, ESHI, and FOCAS Research Institutes for the use of facilities and the support of technical staff and research support office staff.

## Author Contributions

**Conceptualization:** Alessandro Cazzolla.

**Data curation:** James F. Curtin.

**Funding acquisition:** Brijesh Tiwari, Alan Casey, James F. Curtin.

**Investigation:** Alessandro Cazzolla.

**Methodology:** Alessandro Cazzolla, Julie Rose Mae Mondala, Alan Casey.

**Project administration:** Alan Casey, James F. Curtin.

**Resources:** Alessandro Cazzolla, Julie Rose Mae Mondala, Janith Wanigasekara.

**Supervision:** Julie Rose Mae Mondala, Brijesh Tiwari, Alan Casey, James F. Curtin.

**Visualization:** Janith Wanigasekara, Joanna Carroll, Noah Daly.

**Writing – original draft:** Alessandro Cazzolla.

**Writing – review & editing:** Alessandro Cazzolla, Julie Rose Mae Mondala, Janith Wanigasekara, James F. Curtin.

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
