## [Decision Letter · Decision Letter 0]

14 Sep 2023

PONE-D-23-22881

RE:Synthesis of cationic liposome nanoparticles using a thin film dispersed hydration and extrusion method

PLOS ONE

Dear Dr. Curtin,

Thank you for submitting your manuscript to PLOS ONE. After careful consideration, we feel that it has merit but does not fully meet PLOS ONE’s publication criteria as it currently stands. Therefore, we invite you to submit a revised version of the manuscript that addresses the points raised during the review process.

We look forward to receiving your revised manuscript.

Kind regards,

Pradeep Kumar, Ph.D.

Academic Editor

PLOS ONE

Journal Requirements:

   "This study was funded by a TU Dublin Research Scholarship and Science Foundation Ireland, 22/FFP-A/9189. The authors wish to thank TU Dublin, ESHI, and FOCAS Research Institutes for the use of facilities and the support of technical staff and research support office staff"

   "TU Dublin Research Scholarship 2022 (AC) www.tudublin.ie

Science Foundation Ireland, 22/FFP-A/9189 (JC) www.sfi.ie

The funders did not and will not have a role in study design, data collection and analysis, decision to publish, or preparation of the manuscript."

6. Please ensure that you refer to Figure 7 in your text as, if accepted, production will need this reference to link the reader to the figure.

8. We note you have not yet provided a protocols.io PDF version of your protocol and/or a protocols.io DOI. When you submit your revision, please provide a PDF version of your protocol as generated by protocols.io (the file will have the protocols.io logo in the upper right corner of the first page) as a Supporting Information file. The filename should be S1_file.pdf, and you should enter “S1 File” into the Description field. Any additional protocols should be numbered S2, S3, and so on. Please also follow the instructions for Supporting Information captions [https://journals.plos.org/plosone/s/supporting-information#loc-captions]. The title in the caption should read: “Step-by-step protocol, also available on protocols.io.”

Please assign your protocol a protocols.io DOI, if you have not already done so, and include the following line in the Materials and Methods section of your manuscript: “The protocol described in this peer-reviewed article is published on protocols.io (https://dx.doi.org/10.17504/protocols.io.[...]) and is included for printing purposes as S1 File.” You should also supply the DOI in the Protocols.io DOI field of the submission form when you submit your revision.

If you have not yet uploaded your protocol to protocols.io, you are invited to use the platform’s protocol entry service [https://www.protocols.io/we-enter-protocols] for doing so, at no charge. Through this service, the team at protocols.io will enter your protocol for you and format it in a way that takes advantage of the platform’s features. When submitting your protocol to the protocol entry service please include the customer code PLOS2022 in the Note field and indicate that your protocol is associated with a PLOS ONE Lab Protocol Submission. You should also include the title and manuscript number of your PLOS ONE submission.

Reviewers' comments:

Reviewer's Responses to Questions

**Comments to the Author**

1. Does the manuscript report a protocol which is of utility to the research community and adds value to the published literature?

Reviewer #1: No

Reviewer #2: Yes

2. Has the protocol been described in sufficient detail?

To answer this question, please click the link to protocols.io in the Materials and Methods section of the manuscript (if a link has been provided) or consult the step-by-step protocol in the Supporting Information files.

The step-by-step protocol should contain sufficient detail for another researcher to be able to reproduce all experiments and analyses.

Reviewer #1: Partly

Reviewer #2: Partly

3. Does the protocol describe a validated method?

Reviewer #1: Yes

Reviewer #2: Yes

4. If the manuscript contains new data, have the authors made this data fully available?

Reviewer #1: Yes

Reviewer #2: Yes

**5. Is the article presented in an intelligible fashion and written in standard English?**

Reviewer #1: Yes

Reviewer #2: Yes

6. Review Comments to the Author

Reviewer #1: The manuscript describes the very standard procedure of liposome formation using hydration method by using AVANTI extrusion kit. Generally, the paper describes standardly used procedures but there are already at least 2 protocols at the protocols.io already existing: https://www.protocols.io/view/liposome-encapsulation-of-hydrophilic-and-hydropho-rm7vzymorlx1/v1;
https://www.protocols.io/view/two-step-protocol-preparation-and-extrusion-of-pho-n2bvj95xlk5w/v3

These two protocols describe the same procedures as this lab protocol and it is hard for me to see any improvements. On the other hand, in the end of this paper, work with 2D cell lines is showed. Liposomes are used in the cell line tests but on the protocol.io I did not find any lab protocol of this usage. Then the additional value of this protocol may be sufficient.

Specific comments:

26-27 An optimised method was developed ...

What is the optimised of your method? Can you add parameters you have optimised? You are using combination of 100 nm a nd 50 nm membrane, specific concentration of liposomes, etc.

37-38 Lipid nanoparticles, particularly liposomes, have been identified as potentially viable and flexible drug vesicles ...

This sounds to me like refering to a new potential material. Liposomes are 60 years old and nearly 30 years already on the market. I think this should be reflected in the introduction.

72-74 According to available evidence, a liposome diameter of less than 200 nm is the optimal size for effective drug administration, particularly when targeting the brain and crossing the blood brain barrier(13)

Here, you are citing your older version of this protocol?

Generally I have problem with the introduction that it states general knowledge about liposomes that is known for decades and in the same time citing very recent articles that are not primary sources of these informations. From your intraduction, liposomes seem to me like a super new potential material that yet needs to be studied.

99-100 dissolved in a fixed amount of chloroform and stirred for 15 minutes above the lipid’s transition temperature, Tc.

What amount of chloroform and at which temperature? In the protocol, the chloroform amount is stated, temperature not. Were these quantities optimised as it is stated in abstract?

119 ... and surface charge.

I would be very careful with connections of Zeta potential and surface charge. This may vary with ionic strength, size (as you even show)

126 - please check italic/normal characters in the equation

Nowhere in the methods I see the procedure for the doxorubicin encapsulation and purification. If there was purification used, the calculation of EE does not make sence as there was doxorubicin removed from the solution

190 - liposomi

193-194 Here and on many other places you overusing the digits in calculation. There is no point is showing that your measured value is 64.087 ± 33.98. Reasonable amount of digits need to be shown in the whole publication.

3.2 Extrusion study passes: The study you are here performing was surely done many times and event the manual from the manufacturer contains this information, for example here: https://www.ncnr.nist.gov/userlab/pdf/E134extruder.pdf

212-214 ... on the other hand, values less than ±30 can result in particle aggregation, flocculation, and precipitation due to the van der Waal forces of attraction which result in physical instability of the colloidal suspension(15).

Here, you are citing the paper in which this information is stated also with citation. It is very popular to give some border for the Zeta potential but from my point of view this is not correct. The colloidal stability is given by DLVO theory by the combination of electrostatic repulsion and van der Walls attraction. If this value exists it is material specific (in this case for liposomes).

218 - 219 ... confirming the effectiveness of the proposed extrusion method ...

Here it seems like a new method.

223 ... surface charge, 26.7 ± 15.6 mV

Please note, Zeta potential is not a surface charge

3.5 Encapsulation % If I understand correctly, you are using Ultracentrifugation and then without the knowlenge about volumes, you are substracting the concentrations. I think this is not correct approach.

The final part about cytotoxicity is not in the protocol. If something has added value considering the previously published protocols it is probably this part.

In the results of cytotoxicity test, the ones with longer time from synthesis is more toxic for the cells. This probably means that the doxorubicin leaked from the liposomes and killing the cells, but still you are stating that "Storage at 4 ºC appears to be the best, with stability maintained over 16 weeks." (line 312).

Figure 1 is 1:1 copy from Functionalized liposomes for targeted breast cancer drug delivery, Nel et al. Bioactive Materials 2023 and it is not adding any information. I would suggest either cite the source or to add some relevant info.

Reviewer #2: Dear Authors,

The review paper entitled “Synthesis of cationic liposome nanoparticles using a thin film dispersed hydration and extrusion method” by Cazzolla et al. is a nice manuscript describing the synthesis of cationic liposomes. Although this reviewer believes the manuscript contains important data, suitable to be published, unfortunately, in terms of writing and structure, the manuscript is not professional and challenges the reader in many points. Due to some of the major concerns of this reviewer that are listed below, the manuscript in its current form does not meet all of the requirements for publishing.

Some of the concerns listed below need to be answered by the authors to better justify the data and thus, make it clearer for the readers before the publication:

1. The language throughout the manuscript causes significant misunderstandings and more importantly misstatements. It needs more care than this journal requires.

2. There are several senences in the manuscript where the punctation marks are missing thus making it really hard to follow.

a. Line 190, the word liposomes was misspelled

b. Lines 233-234, what does the authors mean by intensity zeta size and number zeta size?

c. The sentence of lines 241-242 needs to be modified.

d. Lines 270-272, “Results in terms of size will be discussed considering the number and not the intensity, to show better the actual increase, if present, of the liposomes size.” needs some simplification.

e. Line 274, what does no encapsulated case mean and in line 277 the author described it as unencapsulated. Please maintain consistency across the manuscript.

f. Line 276, the text in the manuscript needs to be written in past tense.

3. In Figure 4B, the term Zeta Size in the graph needs to change to Size.

4. The text in the manuscript says Figures 5A and 5B, whereas in the actual figures, ther is no A and B in Figure 5.

5. The text says Figures 6A, 6B and 6C, whereas the actual figures are 7A, 7B and 7C.

6. There are several manuscipts describing the synthesis of cationic liposomes, what makes this one unique. The novelty needs to be discussed further in detail.

7. Use of cationic lipids comes with its own challenges and toxicities. The potential toxicities involved need to be mentioned in the introduction.

8. Furhtermore, the stability of the synsthesized liposomes are attributed to the highly positive zeta potential. Longer shelf-life does not make the liposome safe. A safety study such as a hemolysis study needs to be performed to evaluate the toxicities on cell membranes.

9. Authors also need to perform drug release study as a part of stability study as well.

10. The authors need to discuss more about the drop in zeta potential from time zero to 1 week old samples. The zeta potential at time zero was 47.90 mV whereas in 1 week old samples stored in cold were 7.90 mV and 18.30 mv for DOTAP and DOTAP-PI respectively.

11. The justification of such high standard errors in the zeta potential needs to be discussed as well.

12. A justification of why Doxorubicin was used to study the encapsulation % needs to be described in the manuscript and not Propidium Iodide.

13. Cytotoxicity need to be performed with freshly prepared liposomes as well.

14. The authors need to clearly discuss why the bigger size impacted the cytotoxicity.

15. Furthermore, the authors need to study the safety of these cationic liposomes in healthy cells in addition to the cancer cells.

Overall, the listed concerns do not decrease the scientific merit of the submitted manuscript but make it hard to follow. Thus, this reviewer believes the manuscript will be suitable to publish only after major changes.

Sincerely yours.

7. PLOS authors have the option to publish the peer review history of their article (what does this mean?). If published, this will include your full peer review and any attached files.

Reviewer #1: No

Reviewer #2: No

---

## [Author Response · Author response to Decision Letter 0]

28 Sep 2023

Monday 28th September 2023

RE: PLOS ONE Decision: Revision required [PONE-D-23-22881]

Dear Dr Kumar,

Many thanks for reviewing our manuscript entitled: Synthesis of cationic liposome nanoparticles using a thin film dispersed hydration and extrusion method. We now enclose an updated manuscript where we have addressed all of the reviewer feedback, and we also enclose an accompanying rebuttal letter responding to each point raised.

We have also removed the funding from the acknowledgements section. Please include the following as our amended funding statement: This study was funded by a TU Dublin Research Scholarship (AC) www.tudublin.ie and by Science Foundation Ireland, 21/FFP-A/9189 (JC) www.sfi.ie.

With best wishes,

James and Alessandro (james.curtin@tudublin.ie and alessandro.cazzolla@tudublin.ie) 

Journal Requirements:

We can confirm that the manuscript meets PLoS One style requirements.

 "This study was funded by a TU Dublin Research Scholarship and Science Foundation Ireland, 22/FFP-A/9189. The authors wish to thank TU Dublin, ESHI, and FOCAS Research Institutes for the use of facilities and the support of technical staff and research support office staff"

 "TU Dublin Research Scholarship 2022 (AC) www.tudublin.ie

Science Foundation Ireland, 22/FFP-A/9189 (JC) www.sfi.ie

The funders did not and will not have a role in study design, data collection and analysis, decision to publish, or preparation of the manuscript."

We have amended the acknowledgments and removed reference to funding agencies in this section. Please include the following as our amended funding statement: This study was funded by a TU Dublin Research Scholarship (AC) www.tudublin.ie and by Science Foundation Ireland, 21/FFP-A/9189 (JC) www.sfi.ie.

We are making no changes to our Data Availability Statement and will finalise this and release the DOI number once the manuscript is approved.

The paragraph on ethics statement has been moved to the methodology section, see page 5, lines 103-111

This is now done, the list of figure and table captions has been added in the manuscript.

6. Please ensure that you refer to Figure 7 in your text as, if accepted, production will need this reference to link the reader to the figure.

This has now been corrected. 

This is now done, the list of supporting information files with captions has been added at the end of the manuscript.

8. We note you have not yet provided a protocols.io PDF version of your protocol and/or a protocols.io DOI. When you submit your revision, please provide a PDF version of your protocol as generated by protocols.io (the file will have the protocols.io logo in the upper right corner of the first page) as a Supporting Information file. The filename should be S1_file.pdf, and you should enter “S1 File” into the Description field. Any additional protocols should be numbered S2, S3, and so on. Please also follow the instructions for Supporting Information captions [https://journals.plos.org/plosone/s/supporting-information#loc-captions]. The title in the caption should read: “Step-by-step protocol, also available on protocols.io.”

The pdf related to the protocol was changed to S1_file as specified by the guideline provided, see page 5, line 101

Please assign your protocol a protocols.io DOI, if you have not already done so, and include the following line in the Materials and Methods section of your manuscript: “The protocol described in this peer-reviewed article is published on protocols.io (https://dx.doi.org/10.17504/protocols.io.[...]) and is included for printing purposes as S1 File.” You should also supply the DOI in the Protocols.io DOI field of the submission form when you submit your revision.

The DOI for the protocol is included in the manuscript. 

If you have not yet uploaded your protocol to protocols.io, you are invited to use the platform’s protocol entry service [https://www.protocols.io/we-enter-protocols] for doing so, at no charge. Through this service, the team at protocols.io will enter your protocol for you and format it in a way that takes advantage of the platform’s features. When submitting your protocol to the protocol entry service please include the customer code PLOS2022 in the Note field and indicate that your protocol is associated with a PLOS ONE Lab Protocol Submission. You should also include the title and manuscript number of your PLOS ONE submission.

The protocol is uploaded in protocols.io. 

Reviewer #1: 

26-27 An optimised method was developed ...

What is the optimised of your method? Can you add parameters you have optimised? You are using combination of 100 nm and 50 nm membrane, specific concentration of liposomes, etc.

Thank you for the suggestion. We have explained better the method, also giving quantities that were used in the synthesis, see page 2, lines 26-31

37-38 Lipid nanoparticles, particularly liposomes, have been identified as potentially viable and flexible drug vesicles ...

This sounds to me like referring to a new potential material. Liposomes are 60 years old and nearly 30 years already on the market. I think this should be reflected in the introduction.

72-74 According to available evidence, a liposome diameter of less than 200 nm is the optimal size for effective drug administration, particularly when targeting the brain and crossing the blood brain barrier (13)

Here, you are citing your older version of this protocol?

Generally I have problem with the introduction that it states general knowledge about liposomes that is known for decades and in the same time citing very recent articles that are not primary sources of these informations. From your introduction, liposomes seem to me like a super new potential material that yet needs to be studied.

This is a good suggestion, we have made a couple of changes in the manuscript to address this. Emphasis regarding the discovery of nanoparticles that occurred about 60 years ago, see page 2 and 3, lines 40-42. The correct article has been cited, see page 4 lines 78

99-100 dissolved in a fixed amount of chloroform and stirred for 15 minutes above the lipid’s transition temperature, Tc.

What amount of chloroform and at which temperature? In the protocol, the chloroform amount is stated, temperature not. Were these quantities optimised as it is stated in abstract?

As suggested the quantities and stirring temperature used were specified in the text, see page 6, lines 119-124

119 ... and surface charge.

I would be very careful with connections of Zeta potential and surface charge. This may vary with ionic strength, size (as you even show)

Thank you for that suggestion. All sentences where the term surface charge was used for the zeta potential have been modified as suggested.

126 - please check italic/normal characters in the equation

Nowhere in the methods I see the procedure for the doxorubicin encapsulation and purification. If there was purification used, the calculation of EE does not make sence as there was doxorubicin removed from the solution

Thank you, we have corrected the equation as suggested. Specified the return to starting volume before measuring absorbance regarding EE of doxorubicin, see page 9, lines 184-186

190 - liposomi

Fixed

193-194 Here and on many other places you overusing the digits in calculation. There is no point is showing that your measured value is 64.087 ± 33.98. Reasonable amount of digits need to be shown in the whole publication.

Thank you, the values regarding size and PdI were changed using 1 decimal place and 2 decimal places, respectively

3.2 Extrusion study passes: The study you are here performing was surely done many times and even the manual from the manufacturer contains this information, for example here: https://www.ncnr.nist.gov/userlab/pdf/E134extruder.pdf

No, the method reported is a modification of the one proposed by Avanti, see page 10, lines 220-222

212-214 ... on the other hand, values less than ±30 can result in particle aggregation, flocculation, and precipitation due to the van der Waal forces of attraction which result in physical instability of the colloidal suspension (15).

Here, you are citing the paper in which this information is stated also with citation. It is very popular to give some border for the Zeta potential but from my point of view this is not correct. The colloidal stability is given by DLVO theory by the combination of electrostatic repulsion and van der Walls attraction. If this value exists it is material specific (in this case for liposomes).

A change was made regarding what the zeta potential represents for us in terms of stability, keeping in mind that there would be other forces to take into account, see page 11, lines 235-237

218 - 219 ... confirming the effectiveness of the proposed extrusion method ...

Here it seems like a new method.

The proposed method has been changed to the amended method, see page 11, line 241.

3.5 Encapsulation % If I understand correctly, you are using Ultracentrifugation and then without the knowledge about volumes, you are subtracting the concentrations. I think this is not correct approach.

Specified the return to starting volume before measuring absorbance regarding EE of doxorubicin

The final part about cytotoxicity is not in the protocol. If something has added value considering the previously published protocols it is probably this part.

In the results of cytotoxicity test, the ones with longer time from synthesis is more toxic for the cells. This probably means that the doxorubicin leaked from the liposomes and killing the cells, but still you are stating that "Storage at 4 ºC appears to be the best, with stability maintained over 16 weeks." (line 312).

Specified how stability tests refer to DOTAP and DOTAP-PI and not doxorubicin.

Figure 1 is 1:1 copy from Functionalized liposomes for targeted breast cancer drug delivery, Nel et al. Bioactive Materials 2023 and it is not adding any information. I would suggest either cite the source or to add some relevant info.

The image was created by taking cues from the reported article, the correct citations will be made in the caption of the figure.

Reviewer #2:

1. The language throughout the manuscript causes significant misunderstandings and more importantly misstatements. It needs more care than this journal requires.

Changes have been made to avoid misunderstandings

2. There are several sentences in the manuscript where the punctation marks are missing thus making it really hard to follow.

a. Line 190, the word liposomes was misspelled

b. Lines 233-234, what does the authors mean by intensity zeta size and number zeta size?

c. The sentence of lines 241-242 needs to be modified.

d. Lines 270-272, “Results in terms of size will be discussed considering the number and not the intensity, to show better the actual increase, if present, of the liposomes size.” needs some simplification.

e. Line 274, what does no encapsulated case mean and in line 277 the author described it as unencapsulated. Please maintain consistency across the manuscript.

f. Line 276, the text in the manuscript needs to be written in past tense.

A better explanation has been provided regarding the paragraph dealing with size in terms of distribution by intensity and number. Line 241-242 has been modified, see line 319. Lines 270-272 has been modified, see line 287-288. All "no encapsulated" has been changed to "unencapsulated"

3. In Figure 4B, the term Zeta Size in the graph needs to change to Size.

4. The text in the manuscript says Figures 5A and 5B, whereas in the actual figures, ther is no A and B in Figure 5.

5. The text says Figures 6A, 6B and 6C, whereas the actual figures are 7A, 7B and 7C.

Figure modified and figure citations fixed

6. There are several manuscipts describing the synthesis of cationic liposomes, what makes this one unique. The novelty needs to be discussed further in detail.

The novelty of the proposed method has been better emphasized, with added commentary on the cell lines used, see page 4, lines 85-89

7. Use of cationic lipids comes with its own challenges and toxicities. The potential toxicities involved need to be mentioned in the introduction.

In accordance with what has been suggested the toxicity of using cationic lipids has been specified in the introduction, see page 3, lines 56-582. 

8. Furthermore, the stability of the synthesized liposomes are attributed to the highly positive zeta potential. Longer shelf-life does not make the liposome safe. A safety study such as a hemolysis study needs to be performed to evaluate the toxicities on cell membranes.

10. The authors need to discuss more about the drop in zeta potential from time zero to 1 week old samples. The zeta potential at time zero was 47.90 mV whereas in 1 week old samples stored in cold were 7.90 mV and 18.30 mv for DOTAP and DOTAP-PI respectively.

15. Furthermore, the authors need to study the safety of these cationic liposomes in healthy cells in addition to the cancer cells.

Subsequent studies will need to be done regarding the decrease in zeta potential from week 0 to week 1, see page 13, lines 283-284. Improved explanation provided regarding cytotoxicity measured as weeks go by, with subsequent studies that may also be done on normal cell lines, see page 15, lines331-337

9. Authors also need to perform drug release study as a part of stability study as well.

Stability test was made on DOTAP and DOTAP-PI no doxorubicin

11. The justification of such high standard errors in the zeta potential needs to be discussed as well.

We amended in the manuscript. See lines 212-213 and 254-255

12. A justification of why Doxorubicin was used to study the encapsulation % needs to be described in the manuscript and not Propidium Iodide.

Reason for use of doxorubicin was specified in the introduction, see page, 4 lines 78-83

13. Cytotoxicity need to be performed with freshly prepared liposomes as well.

Emphasis regarding cytotoxicity performed on the three cell lines, specifying that they were performed with freshly made liposomes, see page 14, line 322. 

14. The authors need to clearly discuss why the bigger size impacted the cytotoxicity.

Improved explanation provided regarding cytotoxicity measured as weeks go by, with subsequent studies that may also be done on normal cell lines, see page 15, lines331-337

---

## [Decision Letter · Decision Letter 1]

9 Nov 2023

PONE-D-23-22881R1RE:Synthesis of cationic liposome nanoparticles using a thin film dispersed hydration and extrusion methodPLOS ONE

Dear Dr. Curtin,

Thank you for submitting your manuscript to PLOS ONE. After careful consideration, we feel that it has merit but does not fully meet PLOS ONE’s publication criteria as it currently stands. Therefore, we invite you to submit a revised version of the manuscript that addresses the points raised during the review process.

Please take note of the comments and concerns raised by Reviewer 1 in response to your revised manuscript.. 

We look forward to receiving your revised manuscript.

Kind regards,

Pradeep Kumar, B.Pharm., M.Pharm., Ph.D.

Academic Editor

PLOS ONE

Reviewers' comments:

Reviewer's Responses to Questions

**Comments to the Author**

1. Does the manuscript report a protocol which is of utility to the research community and adds value to the published literature?

Reviewer #1: No

Reviewer #2: Yes

2. Has the protocol been described in sufficient detail?

To answer this question, please click the link to protocols.io in the Materials and Methods section of the manuscript (if a link has been provided) or consult the step-by-step protocol in the Supporting Information files.

The step-by-step protocol should contain sufficient detail for another researcher to be able to reproduce all experiments and analyses.

Reviewer #1: Partly

Reviewer #2: Yes

3. Does the protocol describe a validated method?

Reviewer #1: Yes

Reviewer #2: Yes

4. If the manuscript contains new data, have the authors made this data fully available?

Reviewer #1: Yes

Reviewer #2: Yes

**5. Is the article presented in an intelligible fashion and written in standard English?**

Reviewer #1: Yes

Reviewer #2: Yes

6. Review Comments to the Author

Reviewer #1: Dear Authors,

I appreciate the minor changes you have performed in order to improve the manuscript, such as better phrasing, clarification about Zeta potential, number of digits, reference in figure 1.

Still I have following major issues that were already stated in first response:

1) Introduction still looks like the liposomes are really new and potential material even after you added 2 lines to the text. Prime example 90-91: "These negative consequences have propelled researchers to create advanced, alternative drug delivery systems that use nanotechnology to transport DOXO (15)"

Here, it again is cited very recent article on DOXO-iron particles which is really something new in development. But the liposomes with DOXO are totally not new. There are dozens of already marketed DOXO - liposome products in market. https://go.drugbank.com/drugs/DB00997. First of them nearly 30 years FDA approved.

I would really suggest to perform deep literature review to have a more relevant introduction.

2) There are at least 2 protocols at protocol.io that are dealing with the same problem. Both of them are better than the one here reported.

https://www.protocols.io/view/two-step-protocol-preparation-and-extrusion-of-pho-n2bvj95xlk5w/v3

https://www.protocols.io/view/liposome-encapsulation-of-hydrophilic-and-hydropho-rm7vzymorlx1/v1

I would suggest to add the part with 2D cell culture and cell viability assay in order to form complete protocol which can help students and researchers to do basic tests in liposomes.

3) There is still not stated how the Doxorubicin was encapsulated. There are numerous methods for this, the most common ones use salt gradients: for example review here: 10.3390/pharmaceutics14020254.

A protocol for this method would be beneficial. Furthermore, the equation for EE calculation is adding concentrations as an extensive value. This is not correct. Using ultracentrifugation, the volumes of all fractions need to be known. There is a plenty of research done on the EE experimental evaluation: https://doi.org/10.1016/j.ijpharm.2017.11.035. Your reported EE around 85 % corresponds to the EE with using salt gradients but it is nowhere stated.

4) 96-98 There is now statement about the novelty of the work. I seriously doubt this statement as the liposomes were intensively tested through last 50 years (for example here: Kitamura et al., 1996, https://aacrjournals.org/cancerres/article/56/17/3986/502500/Intrathecal-Chemotherapy-with-1-3-Bis-2) but if this publication should serve as a protocol it is not necessary to have original research.

Reviewer #2: Dear Authors,

The revised paper entitled “Synthesis of cationic liposome nanoparticles using a thin film dispersed hydration and extrusion method” by Cazzolla et al. is a nice manuscript describing the synthesis of cationic liposomes. The reviewer believes the comments and concers were answered appropriately.

Overall, this reviewer believes the manuscript is suitable to publish the revised manuscript.

Sincerely yours.

7. PLOS authors have the option to publish the peer review history of their article (what does this mean?). If published, this will include your full peer review and any attached files.

Reviewer #1: No

Reviewer #2: No

---

## [Author Response · Author response to Decision Letter 1]

22 Dec 2023

Response to Reviewer 1

I appreciate the minor changes you have performed in order to improve the manuscript, such as better phrasing, clarification about Zeta potential, number of digits, reference in figure 1. 

We thank the reviewer for the agreeing that the changes made improved the manuscript. We will now address the other issues that have been since raised below. 

Still I have following major issues that were already stated in first response:

1) Introduction still looks like the liposomes are really new and potential material even after you added 2 lines to the text. Prime example 90-91: "These negative consequences have propelled researchers to create advanced, alternative drug delivery systems that use nanotechnology to transport DOXO (15)"

Here, it again is cited very recent article on DOXO-iron particles which is really something new in development. But the liposomes with DOXO are totally not new. There are dozens of already marketed DOXO - liposome products in market. https://go.drugbank.com/drugs/DB00997. First of them nearly 30 years FDA approved.

I would really suggest to perform deep literature review to have a more relevant introduction. 

We thank the reviewer for the suggestions. We have extensively rewritten the introduction, highlighting the origins of liposome research, recent advances in drug delivery systems including clinically approved examples of this, and setting the context for this method to accelerate drug development and delivery systems in research labs worldwide. These changes can be seen in the manuscript on lines 43-48, lines 92 to 98 and lines 103 to 105. 

2) There are at least 2 protocols at protocol.io that are dealing with the same problem. Both of them are better than the one here reported.

https://www.protocols.io/view/two-step-protocol-preparation-and-extrusion-of-pho-n2bvj95xlk5w/v3

https://www.protocols.io/view/liposome-encapsulation-of-hydrophilic-and-hydropho-rm7vzymorlx1/v1

We thank the reviewer for highlighting these two protocols. One method was developed previously by our laboratory, but since then we have improved it, for example we have adjusted the quantities of cholesterol and lipids to make the extrusion more straightforward, and we can confirm that our new method reported in this manuscript is a more robust method for drug delivery studies. The other method is a two step protocol for preparing and extruding liposomes. The method differs from ours in a number of ways including the liposome composition. The authors do not include cholesterol. We have found that cholesterol is important for liposome stability and biocompatibility reasons. Moreover, the temperature used by these authors in their protocol is always 60 �C. This is higher than the temperatures we typically use. However, we have not directly compared this method with ours so we are unable to say which one is better. 

I would suggest to add the part with 2D cell culture and cell viability assay in order to form complete protocol which can help students and researchers to do basic tests in liposomes. 

Thank you very much for the suggestion. In the PLoS Lab Protocol, we included a step by step protocol for the synthesis of cationic liposomes in protocols.io. We provided additional experiments and results to provide context in the accompanying PLoS Lab Protocol manuscript. Regarding 2D cell culture and cell viability assays, we believe that these experiments fall into that second category, contextualising the protocol with applications and expected results, rather than being a core part of the actual synthesis protocol. 

3) There is still not stated how the Doxorubicin was encapsulated. There are numerous methods for this, the most common ones use salt gradients: for example review here: 10.3390/pharmaceutics14020254.

A protocol for this method would be beneficial. Furthermore, the equation for EE calculation is adding concentrations as an extensive value. This is not correct. Using ultracentrifugation, the volumes of all fractions need to be known. There is a plenty of research done on the EE experimental evaluation: https://doi.org/10.1016/j.ijpharm.2017.11.035. Your reported EE around 85 % corresponds to the EE with using salt gradients but it is nowhere stated. 

Thank you very much for the suggestion about doxorubicin encapsulation. We have provided this information in the methods section (please see lines 148 and 149). 

Thank you very much for the comment on EE. We have clarified the method (please see line 212-215). The EE changed from 84.8% to 81%, and this is been updated in line 36 of the Abstract, Table 1, line 367 and line 398 and 399 of the Expected Results and Discussion section. 

4) 96-98 There is now statement about the novelty of the work. I seriously doubt this statement as the liposomes were intensively tested through last 50 years (for example here: Kitamura et al., 1996, https://aacrjournals.org/cancerres/article/56/17/3986/502500/Intrathecal-Chemotherapy-with-1-3-Bis-2) but if this publication should serve as a protocol it is not necessary to have original research. 

We thank the reviewer for the comment. We have clarified this statement about the work carried out in glioblastoma cell lines (please see line 110-111).

---

## [Editor Report · Decision Letter 2]

27 Feb 2024

RE:Synthesis of cationic liposome nanoparticles using a thin film dispersed hydration and extrusion method

PONE-D-23-22881R2

Dear Dr. Curtin,

We’re pleased to inform you that your manuscript has been judged scientifically suitable for publication and will be formally accepted for publication once it meets all outstanding technical requirements.

Kind regards,

Pradeep Kumar, Ph.D.

Academic Editor

PLOS ONE

---

## [Editor Report · Acceptance letter]

26 Mar 2024

PONE-D-23-22881R2 

PLOS ONE

Dear Dr. Curtin, 

I'm pleased to inform you that your manuscript has been deemed suitable for publication in PLOS ONE. Congratulations! Your manuscript is now being handed over to our production team.

Kind regards, 

on behalf of

Prof. Pradeep Kumar 

Academic Editor

PLOS ONE